# Looped Transformers for Length Generalization

**Ying Fan[1], Yilun Du[2], Kannan Ramchandran[3], Kangwook Lee[1]**
[1]University of Wisconsin-Madison    [2]Massachusetts Institute of Technology    [3]UC Berkeley

## Abstract

Recent work has shown that Transformers trained from scratch can successfully solve various arithmetic and algorithmic tasks, such as adding numbers and computing parity. While these Transformers generalize well on unseen inputs of the same length, they struggle with length generalization, i.e., handling inputs of unseen lengths. In this work, we demonstrate that looped Transformers with an *adaptive number of steps* significantly improve length generalization. We focus on tasks with a known iterative solution, involving multiple iterations of a RASP-L operation—a length-generalizable operation that can be expressed by a finite-sized Transformer. We train looped Transformers using our proposed learning algorithm and observe that they learn highly length-generalizable solutions for various tasks.

## 1 Introduction

Most algorithmic tasks such as coding, writing mathematical proofs, and reasoning are defined with inputs of variable *length*. The length of an input often correlates with the difficulty of the problem instance. For example, the longer the input, the more difficult the problem tends to be. We say a model perfectly *length-generalizes* if it can solve an algorithmic task on inputs of any length, even if it was only trained on data with inputs up to a finite length [2]. Generally, it is hard to expect models to be trained on inputs with all possible lengths, and we need to rely on length generalization. Also, if a model can length-generalize, it means the model has truly learned the correct algorithmic solution to the task, not just a spurious solution that works only for a certain range of input lengths.

Recently, many works on Large Language Models (LLMs) have shown that we can get more powerful AI models by scaling both compute and data at training time. This scaling approach has indeed succeeded in improving accuracies on various benchmarks. However, even the largest and latest LLMs like [1] which have been trained on much of the existing text on the Internet, still struggle with length generalization [35, 2, 21]. One possible cause is the particular computing model. LLMs are built based mostly on the Transformer architecture [32]. While Transformers can accept a variable length of inputs (that can be processed in parallel), they usually have a fixed depth. This might be sufficient for certain tasks, but not always.

To learn a model that can effectively generalize to longer problems, it is important to consider architectures that can adaptively adjust the computational budget to the difficulty of the tasks [2, 12, 13]. One approach to achieve this is to explicitly generate intermediate output tokens, similar to writing down a scratchpad, which improves LLMs' capability for solving harder problems[25]. In theory, LLMs may generate more scratchpad tokens representing intermediate computation when solving a more difficult task, indicating that they can allocate elastic computation according to the length and difficulty of the given instance. This approach can be learned by explicitly training a model on data with intermediate computation steps [23, 9]. Alternatively, it can be achieved via Chain-of-Thought (CoT) reasoning with few-shot examples [33] or even in a zero-shot manner [20]. Notice that these approaches still use fixed-depth models. While these approaches help solve more complex reasoning tasks, they are still far from achieving near-perfect length generalization for simple algorithmic tasks. For instance, Lee et al. applied CoT for arithmetic tasks but observed that Transformers cannot length generalize even for simple addition tasks [21].

38th Conference on Neural Information Processing Systems (NeurIPS 2024).

Recently, there has been growing interest in using recurrent architectures for reasoning [10, 3, 5, 36]. Unlike standard RNN-type architectures that process different parts of the input sequence incrementally, one can consider a recurrent architecture that processes the entire input sequence multiple times. This architecture passes the intermediate processing output to the next iteration's input, possibly along with the original input. In particular, if the base model used in each iteration is a Transformer, this model is called a Looped Transformer [36].

Looped Transformer can naturally break the limitation of the fixed depth in the standard Transformer architecture: *One can adjust the number of looped steps based on the computational complexity of the underlying algorithmic solution.* Consider a problem set with the following properties: 1) The problems can be solved by a loop of one RASP-L [37] program[1], i.e., each step in the loop can be performed by a decoder-only Transformer with a fixed depth; 2) The number of steps needed in the loop depends on the problem's complexity, i.e., more difficult problems could potentially require more steps to solve. Under the length generalization scheme, we consider the number of steps depending on the problem length, and define this problem set as $n$-RASP-L problems. For $n$-RASP-L problems, if we can learn these length-independent steps, we can utilize an adaptive number of steps to achieve length generalization.

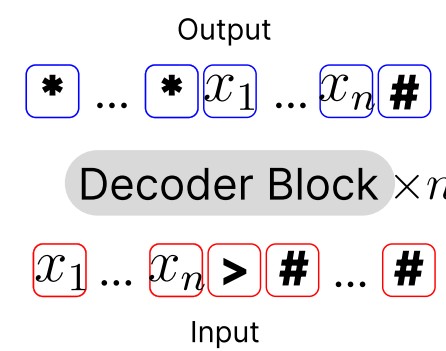

Figure 1: **Method Overview.** During training, we supervise the output of the model to match the target data only after a certain number of steps of applying the same decoder block, helping the model learn intermediate steps that can be reused and can handle input of arbitrary lengths. All grey blocks share the same parameters. Examples are from the Copy task with $n$ symbols. "#" indicates EOS, "*" indicates ignored output, and ">" indicates the end of the query (EOQ).

Inspired by this observation, we study training Looped Transformers models for length generalization. Specifically, we consider a training setup where we do not require any intermediate supervision data (such as reasoning steps or scratchpad). We only assume access to end-to-end supervision (input and output) and the number of steps needed. Depending on the number of steps, we iteratively apply the same decoder block and then decode the final answer; See Figure 1 for illustration. At inference time, the model could either decide when to stop with predefined stopping criteria or stop when reaching the ground-truth number of steps. Empirically, we show that looped Transformers with an adaptive number of steps can successfully length-generalize to longer lengths simply by appropriately adapting the number of loops at inference time, indicating that our approach encourages the model to implicitly learn the necessary steps to solve a task.

Our contributions can be summarized as follows: **(1)** We first formally define $n$-RASP-L problems, and provide examples of $n$-RASP-L solutions to the Copy, Parity, and Addition tasks (Section 2); **(2)** We propose to learn $n$-RASP-L problems with Looped Transformers where we supervise the final answer in a step-dependent way, enabling us to use an adaptive number of steps depending on the problem complexity (Section 3); **(3)** Empirically, we show that our proposed method outperforms the baseline approaches in terms of length generalization performance (Section 5). Due to lack of space, we present the background introduction of RASP-L, next-token prediction and full-answer prediction in Section A, related work in Section 4, and full experimental results in Section C in the appendix.

## 2 $n$-**RASP-L**

RASP-L programs [37] do not allow loops. If we consider the next-token prediction (NTP) scheme, it means that we need to find the same RASP-L program (which can be represented with a fixed-depth decoder-only Transformer) to predict the next token given any possible prefix in the answer sequence. Such solutions might not always exist for all problems: there is no known RASP-L program for addition, parity, and copy under the NTP scheme [37]. On the other hand, architectures such as the Looped Transformer have external loops embedded in the architecture which naturally provides adaptive depth. Thus, a natural question is: what kind of algorithmic tasks can we represent with a decoder-only Transformer in a loop? Specifically, what if we also allow the number of iterations to

---

[1]Here we consider a more general way to loop, i.e., predicting all missing tokens at the end of the loop, not necessarily in the way of predicting the single next token at a time. See more discussions in Section A.2.

explicitly depend on the input length, say $n$? Moreover, what if we are not constrained by the NTP scheme, but a more general full-answer prediction (FAP) scheme?

Inspired by these questions, we define the following class of algorithmic tasks:

**Definition 2.1** ($n$-RASP-L). *A program $P$ is called an $n$-RASP-L program if (1) there exist $T(n)$ :* $\mathbb{N} \to \mathbb{N}$, *and (2) $P$ can be decomposed to a sequential application of $P'$ for $T(n)$ steps.*

We show that $n$-digit addition, $n$-bit parity, copying $n$ symbols indeed have $n$-RASP-L solutions. For the parity task, $P'_{\text{parity}}$ is to shift the input sequence to the right by 1 and calculate XOR of the answer sequence and the input sequence; For the copy task, $P'_{\text{copy}}$ is to shift the input sequence to the right by 1; For the addition task $P'_{\text{addition}}$ is to calculate the XOR of two sequences and shift the results to the right by 1 position as the partial answer, and calculate the AND of two sequences as the carry-on sequence[2]. See Figure 4, Proposition B.1, B.2, B.3 and Listings 1, 2, 3 for details.

# 3 Learning $n$-RASP-L problems with looped Transformers

We present a novel framework for length generalization: In the absence of ground truth CoT data/intermediate output, we propose to leverage the inherent structure of the problem with the help of "knowing when to stop". We present the setup for training data in Section 3.1, the model architecture and training algorithm in Section 3.2, and the inference algorithm in Section 3.3.

## 3.1 End-to-end supervised data without intermediate step supervision

We consider the following settings: (1) There exists an $n$-RASP-L program that solves the given task. (2) Training data consists only of $(x, y)$ pairs, but not intermediate steps. That is, we do not have access to $P'(x), P'(P'(x)), \ldots$. (3) $T(n)$, i.e., the ground truth number of iterations to solve the problem (with some $P'$) is available in the training data[3]. (4) The length $n$ is diversely distributed in the dataset, e.g., $n \in \{1, \ldots, n_{\max}\}$ where $n_{\max}$ is the maximum number of lengths in the dataset; The ground truth number of steps needed $T(n)$ is also diversely distributed in the dataset, e.g., $T(n) \in \{1, , \ldots, T(n_{\max})\}$ where $T(n_{\max})$ is the maximum number of steps in the dataset[4].

## 3.2 Looped training with step supervision

### 3.2.1 Architecture of the looped Transformers

We present the model architecture in Figure 1 with following key characteristics. (1) Recurrence: The Looped Transformer is recurrent (like [15] but with decoder-only structure): We reuse the same decoder block for a number of steps, where each block consists of a certain number of layers. We can adjust the number of looped steps at will. (2) Input injection: For each step, the input embeddings are added to the output embeddings of the previous step as the input of the current step, preventing information loss with improved performance [3, 36]. (3) Positional embedding: There is no positional encoding in the RASP-L operations [37]. To follow our $n$-RASP-L assumption and test the effect of the looped training, we use NoPE [19] to avoid the impact from different positional embeddings[5].

### 3.2.2 Training

Given a dataset $D = \{(\{(x_l)_{l=1}^{L_i}\}_i, \{(y_l)_{l=1}^{L_i}\}_i, T_i, L_i)\}_{i=1}^{N}$, where $\{(\{(x_l)_{l=1}^{L_i}\}_i$ is the input with $L_i$ tokens, $\{(y_l)_{l=1}^{L_i}\}_i$ is the output with $L_i$ tokens, and $T_i$ is ground truth number of steps of sample $i$. We aim to learn the transformer model $M_\theta$[6] by minimizing the following loss:

$$\mathbb{E}_D[\mathcal{L}\left(f_{T_i}(M_\theta, \{(\{(x_l)_{l=1}^{L_i}\}_i), \{(y_l)_{l=1}^{L_i}\}_i\right)], \tag{1}$$

where $\mathcal{L}$ is the cross entropy loss and $f_{T_i}(M_\theta, \{(\{(x_l)_{l=1}^{L_i}\}_i) = \underbrace{M_\theta(M_\theta(\cdots M_\theta}_{T_i \text{ Recursions}}(\{(\{(x_l)_{l=1}^{L_i}\}_i)))$.

---

[2]Here we omit the pre-processing and post-processing steps like handling EOS ("#") and EOQ (">") tokens which can be done by fixed-depth attention layers outside of the loop.

[3]This assumption is to provide supervision for when to stop during training; for inference, we can either use the ground truth steps or leverage the confidence of the output as a stopping criterion (see Section 3.3 for details.)

[4]The length of the problem is not necessarily the same as the actual length of the input due to EOS and EOQ tokens; see Section C.1.1 for the definition of the length of the specific tasks.

[5]NoPE is shown to inherently learn to use relative positional embeddings in practice [19].

[6]$M_\theta$ only handles the embedding space and we use greedy decoding to get the decoded output.

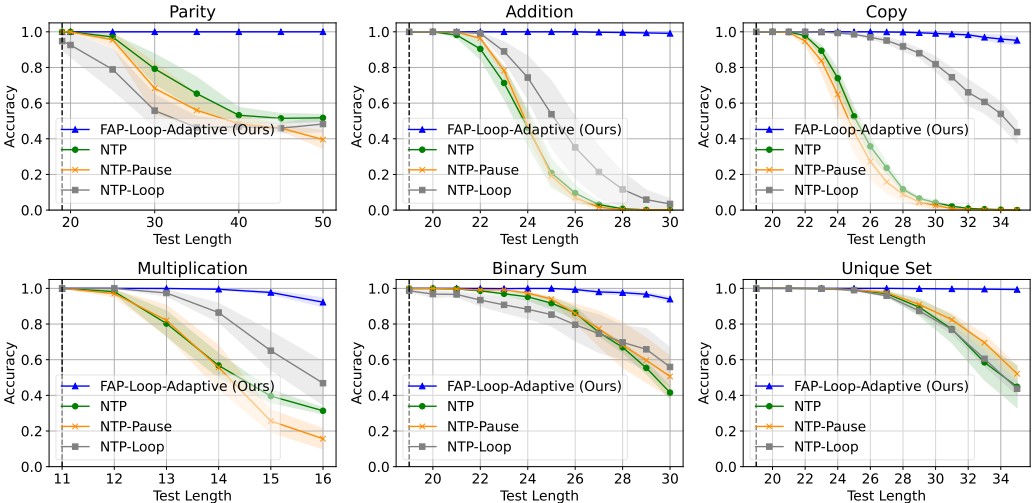

Figure 2: **Length Generalization Performance.** Our looped Transformer model with adaptive depth generalized better than NTP methods across studied tasks, including the variants with pause tokens and weight-tied layers. The vertical dashed line indicates the maximum training length. NTP indicates vanilla next-token prediction; NTP-Pause indicates next-token prediction with pause tokens; NTP-Loop indicates next-token prediction with a fixed number of weight-tied layers.

### 3.3 Adaptive inference

Looped Transformers can use adaptive depth at inference time, so we need certain rules to decide when to stop. We consider two rules: 1) *Oracle*: We can assume that the number of steps needed is given; 2) *Maximum confidence*: We can use confidence base rules to decide when to stop, i.e., stop when we are confident about the output at the current step. More specifically, for 2), given a test sequence $\{(\{(x_l)_{l=1}^L\}$ and a trained model $M_\theta$, we can get the number of steps $T$ from Equation (2):

$$T = \arg\max_{t\in[1,T_{\max}]}\mathcal{L}\left(f_t(M_\theta, \{(\{(x_l)_{l=1}^L\}), \{(\hat{y}_l)_{l=1}^L\}^t\right), \tag{2}$$

where $\{(\hat{y}_l)_{l=1}^L\}^t$ is the decoded sequence from $f_t(M_\theta, \{(\{(x_l)_{l=1}^L\})$ at step $t$, $T_{\max}$ is a threshold for the maximum number of steps.

## 4 Related work

**Positional embedding for length generalization.** Positional embeddings have been shown to greatly affect Transformers' ability to generalize to longer lengths [27, 19, 29, 31, 22, 7, 30, 24, 16]. By designing positional embedding schemes that better capture relative positional information with techniques such as randomization and functional representations, researchers have made significant progress in improving length generalization. Especially, [7] and [24] use tailor-made positional embeddings for some arithmetic problems without potential generality. [7] This direction is orthogonal to our work since there is no positional encoding in RASP-L operations. We choose no positional embedding in our experiments, but other positional embeddings could further be synergistically applied with our approach. However, they might not be expressed as RASP-L operations. We leave further investigation with different positional embeddings to future work.

**RNNs and Chomsky Hierarchy.** [11] conduct an extensive empirical study to investigate the limits of the generalization performance of different neural network structures, demonstrate that grouping tasks according to the Chomsky hierarchy allows forecasting whether certain architectures will be able to generalize to out-of-distribution inputs. Their results show that RNNs and Transformers fail to generalize on non-regular tasks, LSTMs can solve regular and counter-language tasks, and only networks augmented with structured memory (such as a stack or memory tape) can successfully generalize on some context-free and context-sensitive tasks. In our paper, the Looped Transformer

---

[7]In [24], they show that models with weight-tied layers (but with a fixed depth) can improve the generalization ability when comparing with the variants of the same positional embedding, but they do not find adaptive depths to be helpful since they do not perform the step-specific training as our method, while the key to our method is to use models with adaptive depths. To also compare with this baseline, we add NTP-Loop in Section C.1.2.

| Method | Encoder/Decoder | Prediction Type | PE | Input Injection | Halting Mechanism |
|--------|-----------------|-----------------|-----|-----------------|-------------------|
| UT | Both | NTP | Yes | No | ACT [6] |
| PonderNet | Both | NTP | Yes | No | Halting node |
| Ours | Decoder-only | FAP | No | Yes | Confidence based or predefined |

Table 1: Comparison between UT, PonderNet, and ours. PE is short for "Positional Embeddings".

architecture also has augmented memory and the recurrent structure but is potentially more powerful since each iteration contains an operation of the whole sequence.

**Universal Transformers and other looped models.** Our method is highly inspired by Universal Transformers (UT) [10], but we introduce several novel modifications to design looped Transformers that are compatible with our $n$-RASP-L assumption. One major architectural innovation is the use of FAP, while all the other prior works are based on NTP. We also only use decoder-only Transformers, which is different from UT and the follow-up work PonderNet [4], which use both encoder and decoder Transformers. In addition to these two critical differences, we do not use any positional encoding, and use a simpler halting mechanism. Moreover, we find input injection useful to further improve the performance (see details in Section C.3). Table 1 summarizes the differences between ours and the previous approaches. Besides architectural differences, we are also the first to show the benefit of using step-dependent supervision for training looped Transformers. Apart from Transformers, [5] study learning recurrent networks to generalize to harder maze problems than seen during training, but with a focus on CNNs.

**Input representation.** Recall that adding two numbers of length $n$ could not be solved by a RASP-L program where the difficulty mainly comes from indexing operations [37]. It could be solved by reformatting the input so that each digit is presented to the model with "index hints" in [37]. Such reformatting enables a simple RASP-L program for addition. Similarly, representing the answer in reversed order also helps because the corresponding RASP-L program gets much shorter, providing a concrete justification of the empirical observation made in [21]. However, such input representations are highly dependent on the specific problems and might not necessarily exist in general.

**COT.** Scratchpad or CoT reasoning [25, 23, 9, 33, 18] is also useful for length generalization as it could simplify the next-token prediction task with intermediate results presented to the input layer. There are also potential drawbacks and limitations to CoT reasoning. First, CoT training data could be hard to collect. Training and inference with pause tokens [17] has been proposed to learn implicit CoT steps without CoT data, but pause tokens only increase horizontal compute, not sequential compute. Second, not all CoT steps are helpful. If CoT steps introduce additional complexity or require operations not easily expressible in RASP-L, then CoT may hinder length generalization, as shown in [37]. Moreover, CoT steps that could convert the next token prediction task to RASP-L programs might not always exist. Besides, CoT is normally constrained to fixed-depth models, while we study a more general and powerful way to use adaptive compute at inference time.

## 5 Experiments

In this section, we evaluate the efficacy of looped Transformers. We consider tasks with $n$-RASP-L solutions presented in Section 2: Parity, Copy, and Addition, together with more tasks like calculating the sum, multiplication, and calculating the unique set. Due to lack of space, we introduce the detailed experimental setup in Section C.1, present length generalization results in Section C.2, ablation studies in Section C.3, and visualize the stopping criterion in Section C.4 in the appendix.

**Looped Transformers help with length generalization.** As shown in Figure 2, our looped model significantly improves the length generalization performance. For example, for Parity, it can generalize to more than 50 digits near perfectly[8] when only trained with up to 20 digits. Moreover, for tasks like addition and copy, where the next token prediction failed when tested on maximum training length $+10$, our looped model can still perform nearly perfectly. All of the models are only trained with a relatively small number of lengths, and the looped model generalizes surprisingly well.

**Variants of NTP could improve generalization but not as effectively as our adaptive-depth model.** Compared with vanilla NTP, we observe that NTP-Loop could lead to improved generalization in tasks like Addition, Copy and Multiplication. Similarly, NTP-pause could introduce slight improvement in Parity and Unique Set. However, they all fall behind compared with our method. Besides, NTP-Loop suffers from lower in-distribution accuracy in Parity, possibly because using a fixed-depth model with weight-tied layers for NTP with all lengths might be too constrained for the task.

---

[8]It still maintains accuracy higher than 0.95 when tested with 100 digits, which is not included in the graph.

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

# A Background

## A.1 RASP-L

A decoder-only Transformer is a type of Transformer architecture that consists of only the decoder part of the original Transformer model introduced by [32], where a causal mask is applied to the attention weights to prevent the model from attending to future tokens.

RASP (Restricted Access Sequence Processing) [34] is a computational model for the Transformer architecture in the form of a programming language. RASP-L [37], where 'L' stands for learnable, is a learnable subset of the RASP language. Some key points about RASP-L are:

- RASP-L programs accept an input sequence and return an output sequence of the same length for an *arbitrary length*, like decoder-only Transformers.

- The core operations in RASP-L include element-wise operations on sequences and a specific type of non-elementwise operation called `kqv`, which simulates a causal attention layer.

- RASP-L has restrictions on the allowed operations to ensure learnability: It does not allow arbitrary index arithmetic, and restricts operations on token indices to order comparisons and computing successor/predecessor.

- RASP-L *does not allow control flow* statements like branching or loops. Programs must be straight-line code, with each line being a call to a core function or another RASP-L program.

In [37], the authors show that algorithmic tasks that can be written as a RASP-L program can be easily learned by a Transformer in a length-generalizable way with next-token prediction. The length-generalizable tasks include counting, finding the mode, copying the input sequence (consisting of unique tokens), and sorting. However, they also showed that for algorithmic tasks whose RASP-L program representation is not known to exist, such as addition, parity, and copying the input sequence, it is hard to learn in a length-generalizable way. In other words, once the Transformer is trained on in-distribution data up to a particular length, it fails to generalize to unseen lengths.

## A.2 Next-token prediction and full-answer prediction

Decoder-only Transformers are naturally convenient for next-token prediction (NTP) which could be efficiently trained in parallel. In [37], their setup and RASP-L solutions are both constrained to predicting the single next token: During training, the full sequence (both the query and the answer) is provided as input and the output is expected to be the shifted sequence. During inference, only the query part is provided, and the model continues to output the next token and append the token to the current sequence until the output token is EOS. The output locations before the end of query (EOQ) sign are ignored. See (a) in Figure 3 for illustration.

On the other hand, we can also consider a more general way of predicting the answer: full-answer prediction (FAP). During both training

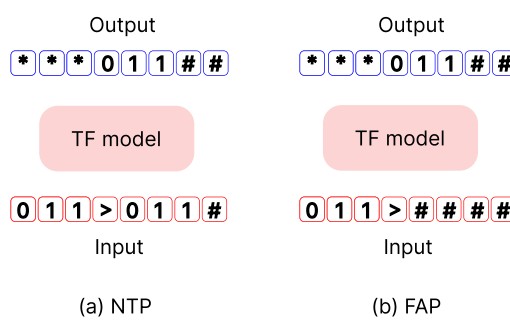

Figure 3: Visualization of the next-token prediction (NTP) and full-answer prediction (FAP) schemes. "#" indicates EOS, "*" indicates ignored output, and ">" indicates the end of the query (EOQ).

and inference time, the input given is just the query part, and the rest of the locations are filled with multiple EOS tokens to keep the input and the output to be the same length. The model is supposed to output the answer with a shifted location, and the output locations before the EOQ sign are ignored; see (b) in Figure 3. Notice that in FAP, the model is not forced to predict token-by-token as NTP. Instead, the model is expected to predict all missing tokens after all internal processing steps.

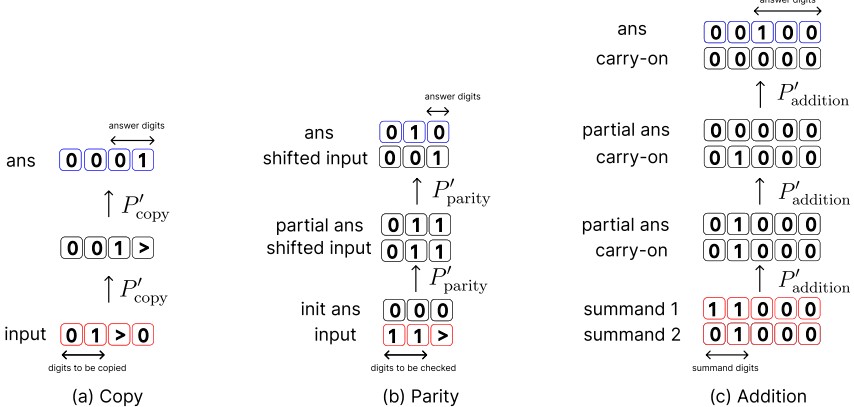

Figure 4: Visualization of the $n$-RASP-L solutions for Copy, Parity, and Addition with $n = 2$. Copy is implemented by $n$ iterations of shifting; Parity is implemented by $n$ iterations of shifting and XOR; Addition is implemented by $n + 1$ iterations of shifted XOR and AND; The inputs are preprocessed.

# B   $n$-RASP-L programs

**Proposition B.1.** *(Parity.) There exists a n-RASP-L program with $T(n) = n$ that solves the n-bit parity check task:*

$$\underbrace{\boxed{x_1}\ \boxed{\ldots}\ \boxed{x_n}}_{n\ tokens}\ \boxed{>}\ \underbrace{\boxed{\#}\ \boxed{\ldots}\ \boxed{\#}}_{n'\ tokens,\ n' \geq 0}\ \Rightarrow\ \underbrace{\boxed{*}\ \boxed{\ldots}\ \boxed{*}}_{n\ tokens}\ \boxed{y}\ \underbrace{\boxed{\#}\ \boxed{\ldots}\ \boxed{\#}}_{n'\ tokens},$$

*where $y$ is the parity check result for the arbitrary binary input sequence $\{x_i\}$.*

*Proof.* See Listing 1 in Appendix B, where the number of steps required in `parity_loop` is $T(n) = n$ for the input query with $n$ bits. □

**Proposition B.2.** *(Copy.) There exists a n-RASP-L program with $T(n) = n$ that solves the n-symbol copy task:*

$$\underbrace{\boxed{x_1}\ \boxed{\ldots}\ \boxed{x_n}}_{n\ tokens}\ \boxed{>}\ \underbrace{\boxed{\#}\ \boxed{\ldots}\ \boxed{\#}}_{n'\ tokens,\ n' \geq n-1}\ \Rightarrow\ \underbrace{\boxed{*}\ \boxed{\ldots}\ \boxed{*}}_{n\ tokens}\ \underbrace{\boxed{x_1}\ \boxed{\ldots}\ \boxed{x_n}}_{n\ tokens}\ \underbrace{\boxed{\#}\ \boxed{\ldots}\ \boxed{\#}}_{n'-n+1\ tokens},$$

*where $\{x_i\}$ is an arbitrary binary input symbols.*

*Proof.* See Listing 2 in Appendix B, where the number of steps required in `copy_loop` is $T(n) = n$ for the input query with $n$ symbols. □

**Proposition B.3.** *(Addition.) There exists a n-RASP-L program with $T(n) = n + 1$ that solves the n-digit addition task:*

$$\underbrace{\boxed{x_1}\ \boxed{\ldots}\ \boxed{x_n}}_{n\ tokens}\ \boxed{+}\ \underbrace{\boxed{y_1}\ \boxed{\ldots}\ \boxed{y_n}}_{n\ tokens}\ \boxed{>}\ \underbrace{\boxed{\#}\ \boxed{\ldots}\ \boxed{\#}}_{n'\ tokens,\ n' \geq n}\ \Rightarrow\ \underbrace{\boxed{*}\ \boxed{\ldots}\ \boxed{*}}_{2n+1\ tokens}\ \underbrace{\boxed{z_1}\ \boxed{\ldots}\ \boxed{z_{n+1}}}_{n+1\ tokens}\ \underbrace{\boxed{\#}\ \boxed{\ldots}\ \boxed{\#}}_{n'-n\ tokens},$$

*where $\{x_i\}$, $\{y_i\}$ are arbitrary binary summands and $\{z_i\}$ is the result of adding $\{x_i\}$ and $\{y_i\}$[9].*

*Proof.* See Listing 3 in Appendix B, where the number of steps required in `addition_loop` is $T(n) = n + 1$ for the input summands with $n$ digits each. □

```
# Input example:  1 1 0 1 > # # #
# Output example: * * * * 1 # # #
# * indicates ignored token, > is EOQ, and # is EOS.

def parity_step(partial_ans_seq, seq):
  # align the last digit with the answer location
  seq = shift_right(seq, 1)
```

---

[9]For simplicity, we include the leading 0's to keep the same length of the output for all possible inputs.

```python
  # calculate XOR
  partial_ans_seq = (partial_ans_seq | seq) \
  & (~(partial_ans_seq & seq))
  return partial_ans_seq, seq

def parity_loop(seq, num_step):
  # get the question in the prompt
  prompt_mask = 1-has_seen(seq, full(seq, EOQ))
  seq = mask(seq, prompt_mask)
  # init answer seq with 0
  partial_ans_seq = full(seq, 0)
  # generate EOS seq after EOQ
  end_seq = where(prompt_mask==1, full(seq, 0), full(seq, EOS))
  # perform parity steps
  for i in range(num_step):
    partial_ans_seq, seq = parity_step(partial_ans_seq, seq)
  # get answer with EOS
  ans_seq = partial_ans_seq
  end_seq = shift_right(end_seq, 1)
  ans_seq = where(end_seq == EOS, end_seq, ans_seq)
  return ans_seq
```

Listing 1: Parity.

```python
# Input example:  0 1 0 1 1 > # # # # # #
# Output example: * * * * * 0 1 0 1 1 # #
# * indicates ignored token, > is EOQ, and # is EOS.

def copy_step(seq, end_seq):
  seq = shift_right(seq, 1)
  end_seq = shift_right(end_seq, 1)
  return seq, end_seq

def copy_loop(seq, num_step):
  # generate EOS seq after EOQ
  end_mask = has_seen(seq, full(seq, EOQ))
  end_seq = where(end_mask==0, full(seq, 0), full(seq, EOS))
  # perform copy steps
  for i in range(num_step):
    seq, end_seq = copy_step(seq, end_seq)
  # get answer with EOS
  seq = where(end_seq == EOS, end_seq, seq)
  return seq
```

Listing 2: Copy.

```python
# Input example:  0 0 1 + 1 1 1 > # # # # # #
# Output example: * * * * * * * * 1 0 0 0 # # #
# * indicates ignored token, > is EOQ, and # is EOS.

def addition_step(seq1, seq2, end_seq):
    end_seq = shift_right(end_seq, 1)
    seq1 = np.array(seq1, dtype = bool)
    seq2 = np.array(seq2, dtype = bool)
    carry_on = seq1 & seq2
    # A XOR B = (A OR B) AND (NOT (A AND B))
    in_place = ((seq1 | seq2) & (~(seq1 & seq2)))
    in_place = shift_right(in_place,1)
    seq1 = np.array(in_place, dtype = int)
    seq2 = np.array(carry_on, dtype = int)
    return seq1, seq2, end_seq

def addition_preprocess(seq):
  # generate EOS seq after EOQ
  end_mask = has_seen(seq, full(seq, EOQ))
```

```
    end_seq = where(end_mask==0, full(seq, 0), full(seq, EOS))
    # generate masks for the first and second summands
    seen_tok0 = has_seen(seq, full(seq, ADD_SIGN))
    seen_tok1 = has_seen(seq, full(seq, EOQ))
    mask1 = ~seen_tok0
    mask2 = seen_tok0 & (~seen_tok1)
    mask2 = mask2 & shift_right(mask2, 1)
    # get the first and second summands
    seq1 = mask(seq, mask1)
    seq2 = mask(seq, mask2)
    # align the first summand with the second
    induct_num1 = cumsum(mask1)
    induct_num2 = cumsum(mask2)
    target_index = firsts(induct_num1, induct_num2, default = 0)
    seq1 = index_select(seq1, target_index)
    seq1 = mask(seq1, mask2)
    return seq1, seq2, end_seq

def addition_loop(seq, num_step):
  seq1, seq2, end_seq = addition_preprocess(seq)
  # perform addition steps
  for i in range(num_step):
    seq1, seq2, end_seq = addition_step(seq1, seq2, end_seq)
  # get answer with EOS
  ans = seq1
  ans = where(end_seq == EOS, end_seq, ans)
  return ans
```

Listing 3: Addition (in forward order).

We also present the RASP-L library functions we use in Listing 4, which is partially taken from [37].

```
import numpy as np

def full(x, const):
    return np.full_like(x, const, dtype=int)

def indices(x):
    return np.arange(len(x), dtype=int)

def select(k, q, pred, causal=True):
    # compute attention matrix
    s = len(k)
    A = np.zeros((s, s), dtype=bool)
    for qi in range(s):
    for kj in (range(qi+1) if causal else range(s)): # k_index <= q_index
     if causal
        A[qi, kj] = pred(k[kj], q[qi])
    return A

def sel_width(A):
    return np.dot(A, np.ones(len(A))).astype(int)

def aggr_mean(A, v, default=0):
    out = np.dot(A, v)
    norm = sel_width(A)
    out = np.divide(out, norm, out=np.full_like(v, default,dtype=float),
    where=(norm != 0))
    return out.astype(int)

def kqv(k, q, v, pred, default=0,):
        return aggr_mean(select(k, q, pred), v, default=default)

def shift_right(x, n, default = 0):
    # shifts sequence x to the right by n positions (other positions
    filled with default)
```

```
        return kqv(indices(x)+n, indices(x), x, equals, default = default)

def where(condition, x_if, y_else):
    # equivalent to np.where(condition, x_if, y_else)
    x_masked = seq_map(x_if, condition, lambda x, m: x if m else 0)
    y_masked = seq_map(y_else, condition, lambda y, m: y if not m else 0)
    return seq_map(x_masked, y_masked, lambda x, y: x if y == 0 else y)

def has_seen(x, queries):
    return kqv(x, queries, full(x, 1), equals, default=0)

def mask(x, bool_mask, mask_val=0):
  # equivalent to x*bool_mask + default*(~bool_mask)
  return where(bool_mask, x, full(x, mask_val))
```

Listing 4: Library functions from [37].

# C  Full Experimental Results

## C.1  Experimental setup

### C.1.1  Tasks

Here we consider tasks with $n$-RASP-L solutions presented in Section 2: Parity, Copy, and Addition, together with more tasks like calculating the sum, multiplication, and calculating the unique set.

**Parity.** Checking the parity of the binary string. Example input: `0 0 0 1 1 > # #`, example output: `* * * * * 0 # #`. We define the length of the problem to be the number of the digits, set $T$ (the number of steps needed) to be the same as the length, and train with length $[1, 20)$.

**Copy (with repeated tokens).** Copying the binary string. Example input: `1 0 1 > # # # #`, example output: `* * * 1 0 1 # #`. We define the length of the problem to be the number of the binary digits to copy, set $T$ to be the same as the problem length, and train with length $[1, 20)$. It has been shown that copy with unique tokens could be easily solved by inductive head [26], but copy with repeated tokens (e.g., binary) does not length-generalize with vanilla NTP training [37].

**Binary Addition.** Performing binary addition of two binary numbers with the same number of digits, and the output has one more digit (without removing leading 0 if it appears). Example input: `1 0 + 1 1 > # # #`, example output: `* * * * * 1 0 1 # #`. We highlight that we do not reverse the output like recent works [21, 24, 38]. We define the length of the problem to be the number of digits to be added, set $T$ to be the same as the problem length, and train with length $[1, 20)$. It has been shown that binary addition without index hint is hard to generalize in vanilla NTP [37].

**Binary Sum.** Calculating the sum of the binary string in the binary form (in reversed order). Example input: `1 0 1 1 > # # # #`, example output: `* * * * 1 1 # # #`. We define the length of the problem to be the number of binary digits to be added, set $T$ to be the same as the problem length, and train with length $[1, 20)$.

**Binary Multiplication.** Multiplying two binary numbers, while the first number has up to two digits. The output is in reversed order and the length is the sum of the lengths of two numbers, without removing leading 0. Example input: `1 1 × 1 1 0 > # # # # #`, example output: `* * * * * * 0 1 0 0 1 0 #`. We define the problem length to be the number of the second digits, and set $T$ to be the product of the lengths of two digits, and train with length $[1, 12)$.

**Unique Set.** Calculating the unique set with the first occurrence order with an alphabet of 50 tokens. Example input: `1 4 2 2 4 3 > # # # # #`, example output: `* * * * * * 1 4 2 3 # #`. We define the length of the problem to be the number of digits to be calculated, set $T$ to be the same as problem length, and train with length $[1, 20)$.

### C.1.2 Baseline methods

**Vanilla NTP.** We use vanilla next-token prediction as one of our baselines, which is referred to as "NTP" in Figure 2. To ensure that the baseline method uses a maximum effective depth comparable to our method during training, we train the transformer model with a depth 20 times the depth of the looped block in our approach.

**NTP with pause tokens.** Training and inference with pause tokens [17] is a way to implicitly learn implicit CoT steps without CoT data by enabling extra compute pathways before outputting the answer in NTP. We use it as a baseline with the same depth as in vanilla NTP which is referred to as "NTP-Pause" in Figure 2. We include a visual illustration of NTP-Pause in Figure 7 in Appendix D.

**NTP with weight-tied layers.** Using weight-tied layers but with a fixed number of overall depths in NTP is also shown to improve the performance in [24]. Here we fix the number of looped steps as 20, use the same depth as the decoder block of our looped model, and train the model with NTP as another baseline which is referred to as "NTP-Loop" in Figure 2.

### C.1.3 Training and evaluation setup

For training, we use a decoder-only Transformer block in GPT-2 architecture [28]. We adopt a curriculum learning strategy for all methods that starts from the smallest length and incrementally increases the length during training till it reaches the maximum length as in [14].

For evaluation, we measure the exact match accuracy for the whole output sequence. For our looped inference, we test two possible stopping criterion discussed in Section 3.3: 1) *Oracle*: Adopt the same rule when generating the dataset as the number of steps to perform, 2) *Maximum confidence*: Run a maximum number of steps, and choose the step using Equation (2)[10]. We report test results from 1) in Section C.2 and C.3, and we also find 2) to be an effective stopping criterion in Section C.4.

Full details of training and evaluation are in Appendix G.

### C.2 Length generalization results

We present the generalization performance on various reasoning tasks in Figure 2.

**Looped Transformers help with length generalization.** Our looped training significantly improves the length generalization performance. For example, for Parity, it can generalize to more than 50 digits near perfectly[11] when only trained with up to 20 digits. Moreover, for tasks like addition and copy, where the next token prediction failed when tested on maximum training length +10, our looped model can still perform nearly perfectly. All of the models are only trained with a relatively small number of lengths, and the looped model generalizes surprisingly well.

**Variants of NTP could improve generalization but not as effectively as our adaptive-depth model.** Compared with vanilla NTP, we observe that NTP-Loop could lead to improved generalization in tasks like Addition, Copy and Multiplication. Similarly, NTP-pause could introduce slight improvement in Parity and Unique Set. However, they all fall behind compared with our method. Besides, NTP-Loop suffers from lower in-distribution accuracy in Parity, possibly because using a fixed-depth model with weight-tied layers for NTP with all lengths might be too constrained for the task.

### C.3 Ablation studies

In Section C.2, we compare with NTP baselines while the efficacy of components in our architecture design remains unclear. In this section, we compare with FAP variants of our model: "FAP-Loop-Adaptive-WO" indicates our method but without input injection; "FAP-Pause" indicates FAP with pause tokens[12]; "FAP" indicates vanilla FAP, without weight-tied layers or adaptive depth.

**Effect of input injection.** We observe that the generalization performance with input injection is generally better than without it, which aligns with the findings in [3, 36]. The effect of input injection is more visible in tasks like Addition, Binary Sum, and Unique Set.

---

[10]Another option is to set a threshold for the cross-entropy loss and stop when the threshold is first met. This will also succeed if the maximum confidence rule works.

[11]It still maintains accuracy higher than 0.95 when tested with 100 digits, which is not included in the graph.

[12]Visual illustration of FAP-Pause is in Figure 8 in Appendix D.

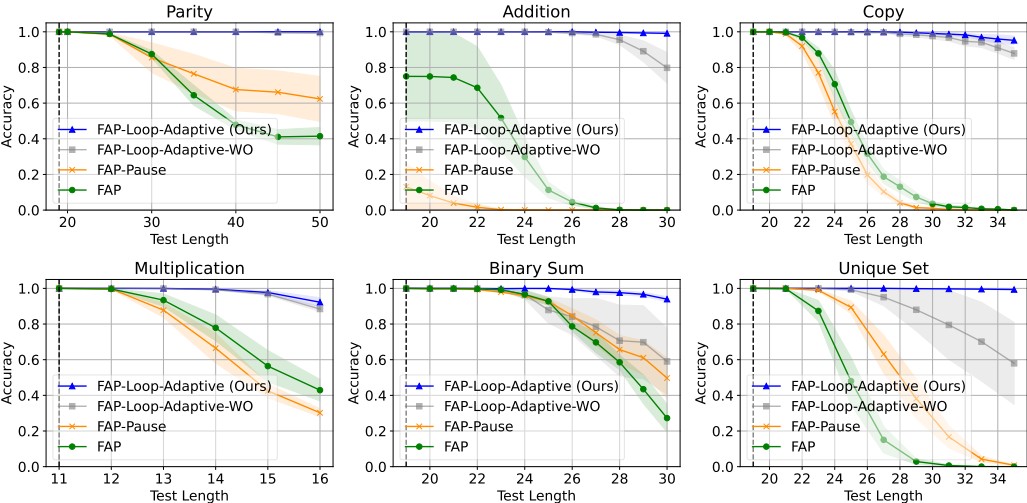

Figure 5: **Ablation study**. Our looped Transformer model with adaptive depth generalized better than FAP variants across studied tasks, including the variant of our method without input injection, and FAP with pause tokens. The vertical dashed line indicates the maximum training length.

**Comparison with pause tokens and vanilla FAP.** We find that training with pause tokens in FAP could boost the generalization performance compared to vanilla FAP, but not as effective as our method with looped steps and adaptive depth. As discussed in [17], pause tokens mainly introduce parallel but not sequential compute, which is less powerful than adaptive depth. Besides, we find worse in-distribution accuracy for both FAP and FAP-Pause in Addition, which mainly comes from the difficulty in training a deep model ($20\times$ the depths of the decoder block used in the looped model) in FAP. It further highlights the importance of supervision with variant depths used in our training.

### C.4 The stopping criterion and visualizations

In this section, we visualize the accuracy and the cross-entropy loss w.r.t. the decoded output in each iterative step across tasks in Figure 6, with the test length to be the maximum length in Figure 2. We also provide more visualizations from other test lengths in Appendix E.

**Convergence in Addition, Copy, Multiplication, and Unique Set.** We notice that for Addition, Copy, Multiplication, and Unique Set, the looped model somehow learns to converge for a certain number of steps after solving the task, even though we do not explicitly train the model to converge. The loss curves for these tasks are also smoother than those without convergence behaviors.

**The maximum confidence stopping criterion chooses the step with near-perfect accuracy.** In Figure 6, the cross-entropy loss reaches the lowest when the generalization performance is near perfect, which indicates the maximum confidence rule chooses the right time to exit. By training with the ground truth number of iterations in the loop, we learn both the length-generalizable iterative steps and when to stop, which is important for looped models.

## D Visualization of using pause tokens in NTP and FAP

We visualize NTP-Pause in Figure 7 and FAP-Pause in Figure 8 respectively, where we add a fixed number of pause tokens (3 in the figures, 20 in our experiments) before outputting the final answer during both training and inference.

## E (More) visualizations of the stopping criterion

Here we present more visualization of the stopping criterion in Figure 9 when tested with different lengths from Section C.4. We can still see similar patterns of convergence in Addition, Copy, Multiplication, and Unique Set. Moreover, the maximum confidence stopping criterion chooses the step with near-perfect accuracy.

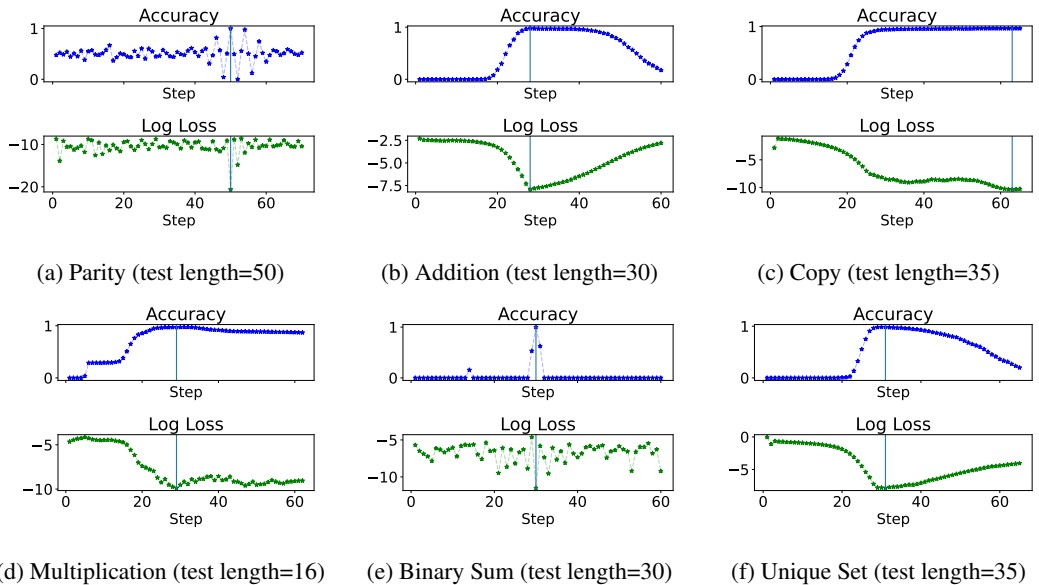

(a) Parity (test length=50)  (b) Addition (test length=30)  (c) Copy (test length=35)

(d) Multiplication (test length=16)  (e) Binary Sum (test length=30)  (f) Unique Set (test length=35)

Figure 6: **Stopping criterion visualizations.** Plot of the stopping criterion. The vertical line indicates the step chosen from Equation (2) within the range shown in the plots. The chosen steps have accuracy $\approx 1$ across tasks.

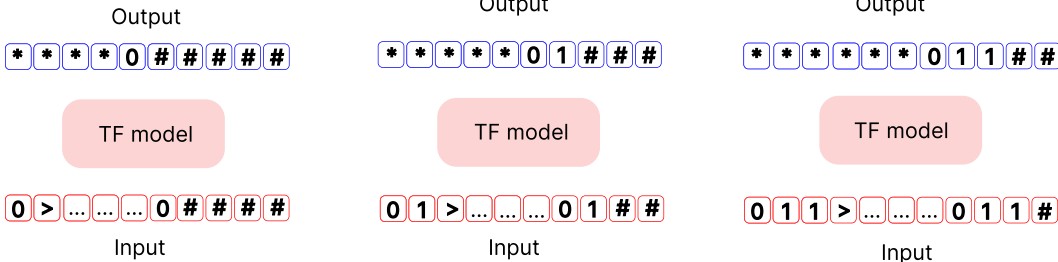

Figure 7: NTP-Pause visualization. Examples are from the Copy task. "..." indicates the pause token.

## F   Inference time complexity

Here we present the inference time complexity for our method, vanilla NTP and vanilla FAP.

Assume that the maximum length of the training set is $n$, the number of steps needed is $T(n)$, and the number of layers in each step is $k$. Assume that NTP and FAP are using a fixed number of layers $C$. And we test on length $n'$.

For the first stopping criterion where we know $a(n')$, our inference time would be $O(ka(n')n'^2)$, and NTP (with KV cache) and FAP will be $O(Cn'^2)$. For the second criterion, we need to specify the maximum number of steps in order to find the step with maximum confidence. So our inference time would be $O(kN'n'^2)$, where $N'$ is the maximum number of steps.

In NTP and FAP, we use some $C \approx kT(n)$ in our experiments such that they use similar compute during training. Our inference time is then slightly longer than NTP with KV cache and FAP since we use more steps than the fixed-depth models.

Moreover, we provide the inference time (in seconds) in Table 2, where we test on length 50 for Parity with batch size 64. Ours (1) and (2) indicate our first and the second stopping criterion respectively.

Table 2: Inference time from Parity.

| Ours (1) | Ours (2) | FAP | FAP-pause | NTP | NTP-pause | NTP (weight-tied) |
|----------|----------|---------|-----------|---------|-----------|-------------------|
| 0.1967s | 0.2190s | 0.1117s | 0.1262s | 0.1229s | 0.1315 | 0.1527s |

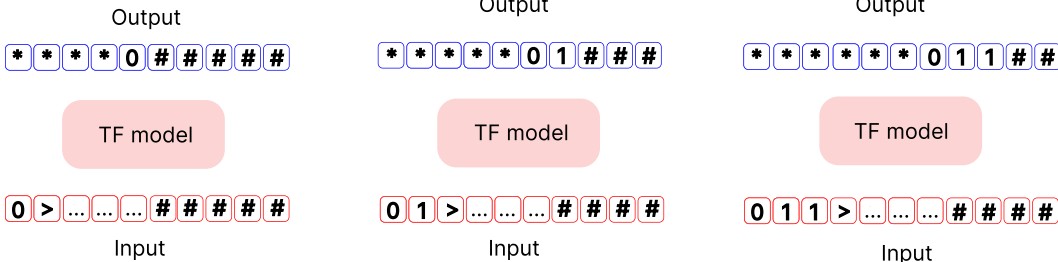

Figure 8: FAP-Pause visualization. Examples are from the Copy task. "..." indicates the pause token.

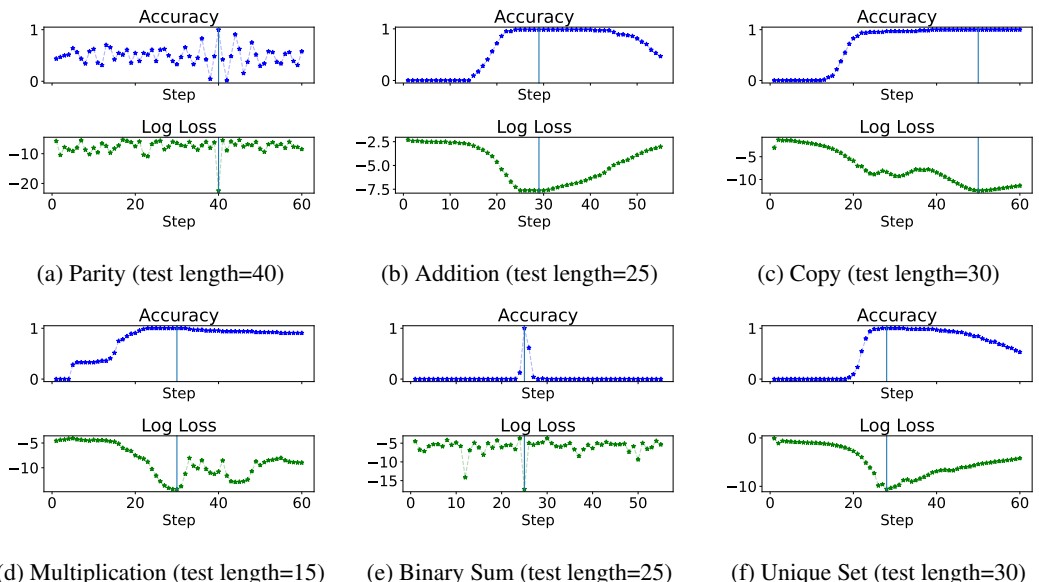

(a) Parity (test length=40)  (b) Addition (test length=25)  (c) Copy (test length=30)

(d) Multiplication (test length=15)  (e) Binary Sum (test length=25)  (f) Unique Set (test length=30)

Figure 9: **Stopping criterion visualizations.** Plot of the stopping criterion. The vertical line indicates the step chosen from Equation (2) within the range shown in the plots. The chosen steps have accuracy $\approx 1$ across tasks.

## G    Experimental details

We use the decoder-only GPT-2 model with NoPE, 8 heads, 256 embedding dimensions as the basic block for the looped iterations with task specific depth in Table 3. We convert the input to the embedding space, perform the loop in the embedding space, and decode the final output after the loop stops. We use a curriculum to gradually increase the maximum training length (see Table 3 for the specific setup for each task). We use AdamW optimizer with a cosine learning rate decay schedule from $10^{-4}$ to 0 after reaching the maximum training length, and train up to 100K gradient steps with batch size 64 for all tasks. For the training distribution, we adopt the online training scheme following [37] where each batch is i.i.d. sampled. Given any length, the probability of each possible character is evenly distributed instead of from a finite train set to avoid over-fitting, and the length is also evenly distributed. For input injection, we use a similar technique as [36] that adds the original input embedding to each looped block as part of the input. For vanilla NTP, we adopt the same training scheme, but trained with autoregressive loss instead. For NTP-Pause and FAP-Pause, we add 20 pause tokens before outputting the final answer. Each training run takes about 4-6h on NVIDIA A100 40 GB GPU, depending on the maximum training lengths of the problems.

For evaluation, we use $100\times$ the batch size number of samples in Figure 2, 5, 6, 9, and report the mean exact match accuracy and standard error from five training runs with different random seeds.

Table 3: Task-specific experimental hyperparameters. "Incremental Interval" denotes the number of training steps between successive increases in the input sequence length.

| Task | Depth of the Decoder Block | Incremental Interval |
|---|---|---|
| Parity | 1 | 1000 |
| Copy | 2 | 1000 |
| Addition | 3 | 1600 |
| Multiplication | 4 | 500 |
| Binary Sum | 3 | 1000 |
| Unique Set | 3 | 1000 |

# H  Limitations and conclusion

Our work has several limitations. Direct looped training could be computationally demanding when the number of looped steps is too large. A possible workaround for more efficient training could be stopping the gradient tracking for earlier steps like [8], but there might be a trade-off in performance and computation. We only train the looped Transformers for a limited number of steps and lengths due to a lack of computing resources. With more diverse training data, the looped model has the potential to generalize to even longer test lengths. We use NoPE for simplicity, and an orthogonal direction is to use more delicate positional embedding to further improve length generalization performance. Moreover, our step-dependent supervision requires the ground-truth number of steps in the training data, which is an additional requirement compared with normal end-to-end training. However, we still require fewer assumptions than CoT training.

In conclusion, we show that $n$-RASP-L problems can be learned by looped Transformer with step-dependent supervision on the final answer, and can be applied with an adaptive number of steps during inference time to improve generalization. Note that $n$-RASP-L, as a challenging algorithmic problem set, could cover more challenging reasoning problems than presented in the paper, and we believe that our method has the potential to generalize to more challenging tasks.

