# OpenReview forum: "Looped Transformers for Length Generalization"
_NeurIPS.cc/2024/Workshop/MATH-AI — MATH-AI 24_

### Official Review · Reviewer_Qpnc · 2024-10-02
**The paper uses looped decoder only transformers (GPT-2 with shared weights across layers) to solve problems of arithmetic**

**Rating:** 7
**Confidence:** 5

**Review:**

The authors propose a universal transformer architecture to solve problems of arithmetic. The transformer has several shared layers, which are iterated through. During inference, the number of iterations can be adaptively increased, to allow the model the generalize to longer sequences than those seen at training.

This is an interesting, and well written paper. My main criticism is that the experiments, and their results,  are scarcely documented in the main. This is sometimes misleading. For instance, the main paper speaks of addition and multiplication, but fails to mention that only the binary cases are considered (this is only stated in Appendix D1). Now, looking at Figure 2, one sees multiplication generalizing from 11 to 16 digits, quite an achievement in base 10, much less so in base 2...

In a short workshop format, I would suggest to focus more on the experiments and their result in the main, and move some of the Rasp-L discussion to the appendix.

---

### Official Review · Reviewer_NecX · 2024-10-03
**Review for Submission49**

**Rating:** 7
**Confidence:** 5

**Review:**

Summary: authors showed that for math task it's possible to train a looped transformer model (i.e. repeating the decoder block n times, where n depends on the input length) with no positional embedding (!) to successfully learn some tasks that a standard transformer cannot do.

Strength:
- This is the first work which achieves a non-trivial length generalization on math problems without using tricks on the positional embeddings (i.e. extending RoPE).
- Authors show that this idea can be applied to all problems not solvable with transformer, but can be solved with a "looped" transformer.
- Results are pretty astonishing.

Weakness/Question:
- Maybe you could combine this with CoT to see if you get higher results on very hard problems?
- Some additional work should be done on stopping criterion.

---

### Official Review · Reviewer_RBbA · 2024-10-06

**Rating:** 8
**Confidence:** 4

**Review:**

This paper studies the performance of looped transformers on length generalization. They are interested in n-RASP-L problems i.e. problems that can be solved by a loop of one RASP-L program. The idea being that the Transformer can learn length-independent steps and we can use an adaptive number of steps in the looped transformer to achieve length generalization. First, they show that n-digit addition, n-bit parity, copying n symbols have n-RASP-L solutions. Then, they show that when using the looped transformer with adaptive stopping time, they get a much stronger length generalization than next token prediction (NTP)  but also other schemes such as using pause tokens, or NTP-loop with  a fixed stopping time.

Overall, I really liked the paper, I think it brings an interesting idea of using a looped transformer to achieve length generalization. This is why I advocate for acceptance of this paper. I have a

What would happen if you have a very deep universal transformer? Universal transformers also have shared parameters and looks equivalent to the loop transformer. The depth may play the role of the number of loops. Would this be equivalent to the fixed loop NTP?


What is the depth of the encoder block in the loop transformer? I think this information is important to put in the main paper.


Where would you position the looped transformers in the list of all the tricks for length generalization? Are the effects similar or complementary to change of the input (index hinting, reverse order of operands, etc.) ? Changes of positional encoding? Chain of Thought? It would be interesting to understand this by making combinations of the tricks with looped transformers with other tricks and analyze the performance differences.


Minor point: the length generalization results may not look very impressive (especially given the state-of-the art results).

I think one weak point of the method is actually coming up with an adaptive inference time. The methods that are proposed are nice but may look a bit hacky. Do you think one could learn this adaptive inference time?

In Figure 2, which adaptive inference time method is used for FAP-Loop-Adaptive?

Lastly, this is a wild question: have you tried your method on problems where there is no n-RASP-L solutions?  Would it still work better than just doing NTP?

---

### Decision · Program_Chairs · 2024-10-07

Accept